# Mechanical Performance and Chloride Penetration of Calcium Sulfoaluminate Concrete in Marine Tidal Zone

**DOI:** 10.3390/ma16072905

**Published:** 2023-04-06

**Authors:** Xudong Tang, Shulin Zhan, Qiang Xu, Kui He

**Affiliations:** 1College of Civil Engineering and Architecture, Zhejiang University, Hangzhou 310058, China; 2Ocean Academy, Zhejiang University, Zhoushan 316021, China

**Keywords:** calcium sulfoaluminate cement, drying–wetting cycles, chloride penetration, marine tidal environment

## Abstract

The enhancement of the durability of sulfoaluminate cement (CSA) in marine environments is of great importance. To this end, an investigation was carried out involving the placement of CSA concrete in the tidal zone of Zhairuoshan Island, Zhoushan, China, and subjected to a 20-month marine tidal exposure test. The comparison was made with ordinary Portland cement (OPC) concrete to evaluate the effectiveness of the former. The test findings indicate that the compressive strength of both types of concrete is reduced by seawater dry-wet cycling, and the porosity of the surface concrete is increased. However, the compressive strength of CSA concrete is observed to be more stable under long-term drying–wetting cycles. When the ettringite in the CSA surface concrete is decomposed due to carbonization and alkalinity reduction, its products will react with Ca^2+^ and SO_4_^2−^ in seawater to regenerate ettringite to fill in the concrete pores, making the concrete strength more stable and hindering chlorine penetration. Furthermore, CSA concrete exhibits a higher capillary absorption capacity than OPC concrete, which results in chloride accumulation on its surface. However, the diffusion capacity of chloride in CSA concrete is significantly lower than that in OPC concrete.

## 1. Introduction

Calcium sulfoaluminate cement (CSA) is a cementitious material produced from raw limestone, bauxite, and calcium sulfate [1]. CSA raw meal contains lower content of limestone than ordinary Portland cement (OPC) raw meal, leading to reduced CO_2_ emissions during the calcination of CSA clinker [2,3,4]. Meanwhile, the calcination temperature required for producing CSA clinker is lower by 100–200 °C compared to the clinkerization process of OPC, resulting in less energy consumption [5,6]. Moreover, the emissions of SO_2_ are nearly negligible when the calcination of CSA clinker is under oxidizing atmosphere. Therefore, CSA cement represents an important aspect of green development in the cement industry.

The main mineral constituents of CSA cement are ye’elimite (C_4_A_3_$\overset{-}{S}$), belite (C_2_S), and gypsum (C$\overset{-}{S}$). Concerning the hydration of CSA cement, ye’elimite reacts with gypsum to produce ettringite (C_3_A·3C$\overset{-}{S}$·32H, AFt) and amorphous aluminum hydroxide (AH_3_) [7]. Meanwhile, monosulfoaluminate (C_3_A·C$\overset{-}{S}$·12H, AFm) forms when the amount of gypsum is insufficient during the hydration of ye’elimite [8,9]. C_2_S reacts with water to form calcium silicate hydrate gel (C-S-H) and portlandite (CH). Subsequently, the reaction between portlandite, aluminum hydroxide, and gypsum results in the formation of ettringite [10]. CSA cement hydrates rapidly and exhibits a short setting time. Most hydration heat evolution occurs between 2–12 h of hydration [11], resulting in high early-age strength and rapid hardening. Additionally, CSA cement has excellent dimensional stability due to the shrinkage compensation effect based on the rapid formation of ettringite [1,12,13].

In the marine environment, the degradation of concrete structures is caused by multiple physical and chemical factors such as various agents in seawater, including sulfate ions, magnesium ions, and chloride ions, etc., carbonation, freeze-thaw damage, the scouring of waves and floating objects and microbial attachment, etc. [14,15,16,17,18]. Concrete structures exposed to the tidal zone are particularly vulnerable to corrosion [19]. The corrosion of steel rebar in concrete, particularly, is a more significant concern because of the frequent cycles of drying and wetting, which offer sufficient amounts of chloride and oxygen [20]. Therefore, the durability of CSA cement in the marine tidal zone, especially its resistance to chloride penetration, is of great importance.

It is widely acknowledged that CSA concrete exhibits superior performance in freeze-thaw environments compared to OPC concrete owing to its lower porosity and the inclusion of more coarse pores [21,22]. It has also been posited that CSA concrete is more resistant to sulfate attack due primarily to the major hydration product, i.e., ettringite does not react with sulfate [9,23]. However, in contrast to OPC concrete, CSA concrete is more vulnerable to carbonation, which results from the absence of portlandite in its hydrated phases. The chloride resistance of CSA concrete has been the subject of debate, with some studies suggesting that it performs better than OPC concrete as assessed through rapid chloride permeability tests [22,24]. An investigation of CSA concrete in a tidal zone also demonstrated its effectiveness in protecting steel bars, which was attributed to the strong self-desiccation of CSA concrete, creating a dry internal micro-environment that hinders chloride penetration into the concrete [25,26]. Nevertheless, other studies have indicated that CSA concrete exhibits low alkalinity and a weak ability to bind chloride, and the faster penetration of chloride within the samples results from this. Additionally, the ability of chloride ions to bind within CSA concrete decreases as the gypsum content in the cement increases, while ettringite is not capable of chemical ion binding. On the other hand, AFm can react with ions and produce Friedel’s salt. Most studies on CSA cement have been conducted on a laboratory scale, further investigation into its practical application in marine environments is necessary. Therefore, the purpose of this study is to conduct further research on the mechanical properties and chloride penetration of CSA concrete in marine tidal zone

## 2. Experimental Procedure

### 2.1. Materials

For this study, the chemical composition presented in Table 1 was used for both 42.5-grade calcium sulfoaluminate cement (CSA) and 42.5-grade ordinary Portland cement (OPC). The coarse aggregates employed were limestone with a size range of 5–25 mm, while medium river sand with a fineness modulus of 2.4 was used as the fine aggregate. The grading curves of fine aggregates and coarse aggregates are shown in Figure 1. The solid superplasticizer with a water reduction rate of 25% was also applied as the water-reducing agent. Further, the boric acid was employed as the retarder for calcium sulfoaluminate cement to get adequate workability. The mix formulation of CSA and OPC concrete is shown in Table 2.

### 2.2. Sample Preparation

To conduct the compression and chloride penetration tests in the marine environment, the 100 × 100 × 100 mm cubic concrete samples were prepared. After the concrete was molded for 6 h, the molds were removed and placed in a standard curing room with a temperature of 20 ± 2 °C and a relative humidity of 98 ± 2% for 28 days. To achieve one-dimensional chloride penetration, all sides of the concrete samples, except one, were coated with epoxy resin after 21 days of curing. The samples were then left to dry for 3 days, followed by applying epoxy resin to one side for a day and an additional 3 days of drying. This process was completed within the 28 days of standard curing; then, the concrete samples were transported to Zhairuoshan Island and placed in the tidal zone for the marine exposure test. Some samples remained in the standard curing room as control samples.

### 2.3. Marine Environment

The samples were placed in the tidal zone at the Material Corrosion Field Observation and Research Station of Zhejiang University, in Zhairuoshan Island, Zhoushan, Zhejiang Province (N 29.95°, E 122.09°) as shown in Figure 2a,b. The island belongs to a subtropical monsoon climate with an annual mean temperature of 16 °C and a salinity of 2.5% in seawater. The tide in this area is irregular semi-diurnal tide with a mean tidal range of 1.95 m and a maximum tidal range of 4.24 m. Meanwhile, the mean interval time of neighboring high and low tides is 10.3 h and 4.5 h, respectively. More details of the atmospheric and water environment of Zhairuoshan Island are shown in Table 3, and the composition of seawater is shown in Table 4 [27,28,29,30].

### 2.4. Methodologies

#### 2.4.1. Compression Test

The compressive strength of the concrete samples was measured in triplicate using a YAW-3000B computer-controlled electro-hydraulic pressure testing machine (produced by Julong, Nanjing, China) with a loading rate of 0.5 MPa/s. The strength values were obtained after a specified period of either standard curing or marine tidal exposure tests.

#### 2.4.2. Rapid Chloride Permeability Test 

The concrete samples, with a thickness of 30 mm, were ground into powder using an HDM-1A concrete grinding machine (produced by Xinyu, China). The grinding process involved grinding every 1 mm of the first 6 mm of the sample’s thickness and every 2 mm for the remaining thickness. To ensure accuracy, powder samples were collected from two identical samples. The collected powders were passed through a 0.16 mm square hole sieve and then stored in sealed bags for the free chloride concentration test. The free chloride concentrations were measured using a rapid chloride ion content analyzer (Model: DY2501-B; Producer: Daejeon, Korea). Prior to measurement, 2 g of powder was dissolved in 20 g of deionized water, and the mixture was shaken vigorously for 10 min and left to stand for 24 h.

#### 2.4.3. Pore Structure Test

To prepare for the mercury intrusion porosimetry test, the surfaces of the concrete samples, ranging from 0–10 mm, were crushed to a particle size of 2–3 mm (with the removal of the coarse aggregates) and preserved in alcohol. The samples were subsequently dried at 60 °C for 2 days and tested using an Auto Pore 9500 injection apparatus (produced by Micromeritics, Norcross, GA, USA).

#### 2.4.4. X-ray Diffraction (XRD) and Thermogravimetric Analysis (TGA)

To conduct X-ray diffraction analysis, the 2 mm-thick concrete surface samples were ground into <0.16 mm powders. The analysis used a D8 advance instrument (Producer: Bruker, Germany) with Cu Kα (λ = 1.54 Å) radiation, operating at 40 kV voltage and 40 mA current in Bragg-Brentano geometry. The samples were scanned in the 2θ range of 5–89° at a scan rate of 0.02°.

For the TGA analysis, the samples were prepared the same way as for the XRD analysis. The analysis was conducted using an SDT Q600 instrument (Producer: TA, New Castle, DE, USA) with a temperature increment of 10 °C/min in a nitrogen atmosphere, ranging from 25 to 1000 °C. After derivation, the DTG curves were obtained, showing peaks corresponding to the decomposition of various hydration phases.

## 3. Results

### 3.1. Compressive Strength

Figure 3 depicts the impact of marine tidal exposure on the compressive strength of concrete samples made with CSA and OPC. The results indicate that both types of concrete exhibit increased compressive strength when cured under standard conditions, with CSA concrete displaying a higher growth rate than OPC concrete. This can be attributed to the continuous hydration of unreacted cement particles, leading to the filling of pores with hydration products and the consequent strengthening of the concrete [31]. However, compressive strength tends to plateau after 12 and 8 months for CSA and OPC concrete, respectively, indicating that the cement hydration process is essentially complete by that time. In contrast, exposure to marine tidal zone leads to a decrease in compressive strength for both types of concrete due to the deterioration caused by drying and wetting cycles on the concrete surface. In the case of CSA concrete, the deterioration may be caused by the carbonation of ettringite due to CO_2_ in the atmosphere or the conversion of ettringite due to alkalinity and temperature. Moreover, the long-term trends in compressive strength development for CSA and OPC concrete under marine tidal exposure differ, with the former showing an increase in strength up to 8 months, followed by stabilization, and the latter exhibiting a peak at 2 months, followed by a gradual decline. These findings suggest that CSA concrete may offer superior performance in terms of compressive strength under marine tidal exposure. This will be explained in further microstructural analysis.

### 3.2. Pore Structure

The pore structures of the concrete surface after marine tidal exposure for 2 months were investigated. The cumulative pore volume and the pore size distribution of the concrete surface are presented in Figure 4a,b. The penetration of chloride into the concrete in a marine tidal environment occurs primarily via capillary absorption along with moisture on the surface and diffusion within the interior. Capillary absorption has a significant impact on chloride penetration into concrete under drying–wetting conditions. Capillary pores in concrete typically range from 10–10,000 nm [32], with those smaller than 20 nm being classified as harmless, those between 20–50 nm as less-harmful, those between 50–200 nm as harmful, and those larger than 200 nm as more-harmful [33,34]. The results presented in Figure 4a show that the porosity of CSA concrete increased from 14.3% to 15.5% after seawater drying–wetting cycles compared with that of concrete under standard curing conditions. Similarly, the porosity of OPC concrete increased from 13.7% under standard curing conditions to 15.6% after marine drying–wetting cycles. The porosity growth rates of CSA concrete and OPC concrete were 8.4% and 13.9%, respectively, indicating that the deterioration rate of CSA concrete is lower than that of OPC concrete under the condition of marine drying–wetting cycles. Moreover, compared with CSA-S, the volume of pores larger than 200 nm in CSA-M increases, and that of pores smaller than 50 nm also increases. Under the drying–wetting cycles of seawater, the surface of CSA concrete gets carbonized and causes the decomposition of ettringite. With the carbonation process, the alkalinity of the concrete pore solution becomes lower, leading to the transformation of ettringite into AFm [35,36]. In addition, the scouring by seawater containing sediment may weaken the bonding strength between aggregate and cement paste, causing the aggregate particles to fall off and increasing the number of pores. Figure 4b shows that the most probable pore diameter of CSA concrete is about 120 nm, while that of OPC concrete is approximately 40 nm. This is attributed to the different cement hydration product systems. The primary hydration product of OPC concrete is C-S-H gel, whereas that of CSA concrete is ettringite. The interlayer pores of C-S-H gel are much smaller. Therefore, the capillary absorption of CSA concrete is stronger during marine drying–wetting cycles. However, compared with CSA concrete, the number of pores larger than 200 nm in OPC concrete increases greatly after marine drying–wetting cycles, which will accelerate the transmission of chloride ions in the concrete.

### 3.3. Chloride Content

The total chloride content includes free chloride, mainly resulting in steel corrosion due to its penetration, accumulation, and bound chloride. Figure 5a,b presents the distribution curves of free chloride content of CSA and OPC concrete samples exposed to marine tidal zone for 2, 4, 8, 12, 16, and 20 months, respectively. The results show that the free chloride contents in CSA and OPC specimens both increase with the increment of exposure time. Besides, it is observed that CSA concrete exhibits a higher concentration of chloride on its surface in comparison to OPC concrete under equivalent exposure times. Compared with OPC concrete, CSA concrete exhibits greater capillary absorption capacity and increased chloride ion penetration into the concrete surface. Furthermore, the growth rate of chloride content of CSA concrete surface at 1 mm decreases gradually with increasing exposure time, eventually stabilizing after 12 months, while that of OPC concrete surface at 1 mm tends to increase. In addition, OPC concrete exhibits a clear convection zone on the surface with a spike occurring at approximately 2 mm, whereas no such convection zone is observed in CSA concrete. During the concrete transition from a wet state to a relatively dry state, surface water evaporates, increasing the relative humidity of the surface layer [37]. However, this evaporation process only affects the surface layer due to the hysteresis effect, with minimal impact on the interior. Consequently, chloride ions are drawn towards the surface of the concrete by capillary absorption through the humidity gradient, while chloride ions on the surface diffuse towards the interior through the concentration gradient. After repeated drying–wetting cycles, a peak of chloride content appears at a specific depth within the concrete. However, in the case of CSA concrete, the capillary absorption of chloride ions from the interior to the surface is much stronger than the diffusion of chloride ions from the surface to the interior during the transition from a wet state to a relatively dry state. As a result, there is no discernible convection zone on the surface of CSA concrete. Additionally, the chloride content within the interior of CSA concrete is significantly lower than that of the surface layer due to the self-desiccation of CSA concrete [26], which creates a low relative humidity environment within the internal concrete. This greatly inhibits the diffusion of chloride ions into the interior of CSA concrete, making it less susceptible to chloride-induced deterioration.

### 3.4. XRD

The X-ray diffraction (XRD) patterns of concrete surface samples from marine tidal exposure at varying durations are presented in Figure 6. The predominant diffraction peaks for ettringite, AFm, Friedel’s salt, and gypsum were observed at 2θ values of 9.1°, 9.8°, 11.2°, and 11.7°, respectively. Table 5 reports the peak intensity and full width at half maximum of the samples. The XRD results demonstrate that the content of ettringite in the concrete surface initially decreases and then increases with increasing marine exposure time. Gypsum was only detected in the 2mon-M and 4mon-M samples, while Friedel’s salt was observed in the 12mon-M sample.

Further, it can be observed from Table 5 that, compared to standard curing, the surface concrete of CSA concrete exhibited a decrease in ettringite content, deterioration in crystallinity, and a concurrent presence of gypsum after undergoing 2 months of marine drying–wetting cycles. As time progressed to 4 months, the ettringite content continued to decline while the gypsum’s crystallinity improved. Subsequently, with continued marine drying–wetting cycles, the ettringite content began to increase, and the crystallinity improved while the presence of gypsum reduced. Moreover, after 12 months, the poorly crystallized Friedel’s salt was detected and subsequently declined with increased marine drying–wetting cycles. The presence of AFm was not found in each XRD pattern, whereas the missing of AFm in the patterns resulted from its inferior crystalline structure.

### 3.5. TGA

The thermogravimetric-differential thermogravimetric (TG-DTG) curves of concrete samples containing calcium sulfoaluminate (CSA) subjected to marine drying–wettingexposure tests for 2, 4, 12, and 20 months were presented in Figure 7. Additionally, Table 6 summarizes the mass loss of each hydration phase. These curves depict the associated dehydration process of the decomposition of ettringite, AFm, Friedel’s salt, and AH_3_, as well as the decarbonation process of calcite. The weight loss at temperatures of 80–110 °C can be attributed to the decomposition of ettringite, while the weight loss at temperatures around 130–140 °C can be assigned to the dehydration of AFm. Furthermore, the presence of AH_3_ is confirmed by the weight loss at approximately 220–270 °C, whereas the decomposition of Friedel’s salt occurred at temperatures between 300–380 °C, and the decomposition of calcite (CaCO_3_) occurred around 600–700 °C.

Observations from the TG-DTG curves of the 2mon-M phase compared to the 2mon-S phase reveal a significant decrease in ettringite content, an increase in calcium carbonate content, and the appearance of an endothermic peak of AFm. With the marine tidal exposure test, it can be observed from the TG-DTG curves as well as the mass loss table that ettringite content gradually decreased until 4 months, while the content of AFm and calcium carbonate gradually increased. This can mainly be attributed to the decomposition of ettringite. After 4 months, ettringite content began to increase, calcium carbonate content began to decrease, and AFm disappeared, as seen in 12mon-M and 20mon-M. The presence of Friedel’s salt was also found in 12mon-M and decreased greatly in 20mon-M. The above phenomenon can be attributed to the gradual weakening of the decomposition of ettringite with exposure time and the decomposed product of ettringite reacting with Ca^2+^ and SO_4_^2+^ in seawater to regenerate ettringite. Additionally, the disappearance of AFm at 12 months may be related to the formation of Friedel’s salt between AFm and chloride ions.

## 4. Discussion

The compressive strength of CSA concrete exhibits a gradual increase with exposure to marine tidal zone until it reaches a state of stability after 12 months. In contrast, the strength of OPC concrete tends to decrease slowly after 2 months. The increase in strength of CSA concrete is primarily attributed to the continuous hydration of unhydrated cement particles, leading to the formation of hydration products such as ettringite and aluminum glue that fill in gaps within the concrete [38]. The porosity of the surface of CSA concrete increases gradually due to the marine drying–wetting cycles, whereby CO_2_ in the atmosphere reacts with ettringite to produce calcite, AH_3,_ and gypsum. The carbonization process causes a decrease in alkalinity within the concrete pore solution, which reduces the stability of ettringite and leads to its conversion into AFm and gypsum [39,40]. Furthermore, the sand and gravel in seawater continuously scour the concrete surface in the tidal zone, reducing the bonding strength between aggregate particles and cement paste, which can result in the formation of pores. Compared to the pores size distribution of OPC concrete, CSA concrete has more pores with sizes ranging from 50–200 nm, which is responsible for the penetration of solution during the drying stage of concrete. As a result, CSA concrete exhibits stronger capillary absorption under the influence of drying–wetting cycles, coupled with weak chloride binding capacity, leading to higher chloride content in the surface layer. However, the drying–wetting cycle has little impact on the relative humidity inside the concrete due to the hysteresis effect, and the self-desiccation effect of CSA concrete maintains a relatively dry internal environment, thereby slowing down the diffusion of chloride. Consequently, the chloride content in the interior of the concrete is significantly lower than that on the surface layer. From a long-term durability perspective, this is a benefit of using CSA concrete in marine environments. Additionally, CSA concrete does not form a convection zone like ordinary Portland cement concrete due to its higher capillary absorption capacity and lower diffusion capacity, resulting in faster stabilization of chloride content on its surface.

Pores larger than 200 nm in OPC concrete are observed to increase significantly after marine drying–wetting cycles, thereby accelerating the transmission of chloride on the concrete surface. This phenomenon is primarily caused by the reaction of SO_4_^2−^ in seawater with the hydration products such as hydrated aluminates in OPC concrete, resulting in the formation of gypsum and ettringite, which in turn lead to an increase in concrete pores and cracks. Moreover, chlorides can bind to the surface of pores, predominantly on C-S-H and other hydrated cement phases, or react with C_3_A to form Friedel’s salt, which can also contribute to the expansion and cracking of concrete over time [41,42].

The adaptability of the expansion process to the strength development of the concrete is the key factor in determining whether the delayed ettringite expansion will adversely affect the concrete. CSA concrete exhibits strong strength and adaptability to the expansion of ettringite. Thus, during the drying–wetting cycles, the decomposition products of ettringite in seawater rich in Ca^2+^ and SO_4_^2−^ can regenerate ettringite and fill the concrete surface without damaging the concrete, thereby retarding the deterioration caused by the drying–wetting cycles. In addition, CSA concrete has a limited ability to bind chloride, and even if Friedel’s salt forms, it is highly unstable in the pore solution system of CSA concrete. The low alkalinity and high sulfate content make Friedel’s salt prone to decomposition [43,44]. Under marine drying–wetting cycles, ettringite, the main hydration product on the surface of CSA concrete, can continue to decompose and regenerate. In the early stage of marine tidal exposure, the degradation of the concrete surface results in a decomposition rate of ettringite much faster than the rate of regeneration and hydration, thereby decreasing the content of ettringite and increasing the content of gypsum. However, as the exposure time continues to increase, the rate of ettringite generation gradually dominates over the decomposition rate, resulting in the increase of ettringite and the disappearance of gypsum in the surface layer of CSA concrete. Overall, CSA concrete exhibits greater stability of compressive strength than OPC concrete under marine drying–wetting cycles and is more resistant to seawater corrosion.

## 5. Conclusions

The compressive strength of CSA concrete and OPC concrete is susceptible to degradation under the marine drying–wetting cycles. However, it is notable that the compressive strength of CSA concrete exhibits greater stability under prolonged marine drying–wetting cycles;The corrosion mechanism of seawater on the two types of concrete varies under the influence of the marine drying–wetting cycles. OPC concrete is vulnerable to expansion and cracking due to the production of ettringite or Friedel’s salt resulting from the reaction of ions in seawater and its hydration products. On the other hand, in CSA concrete, carbonization and low alkalinity result in the decomposition of ettringite, which then reacts with seawater ions to generate ettringite suitable for its strength development;The penetration mechanism of chloride in the two types of concrete is distinct under the influence of marine drying–wetting cycles. The capillary absorption on the surface of CSA concrete is more pronounced, leading to chloride accumulation, while the interior remains relatively dry, resulting in slower chloride ion diffusion. Additionally, there is no convection zone in CSA concrete.

In the long-term service of concrete in the marine environment, especially in the tidal zone, stable strength and relatively low chloride content are very important. This research reveals the potential of using CSA cement in a marine environment.

## Figures and Tables

**Figure 1 materials-16-02905-f001:**
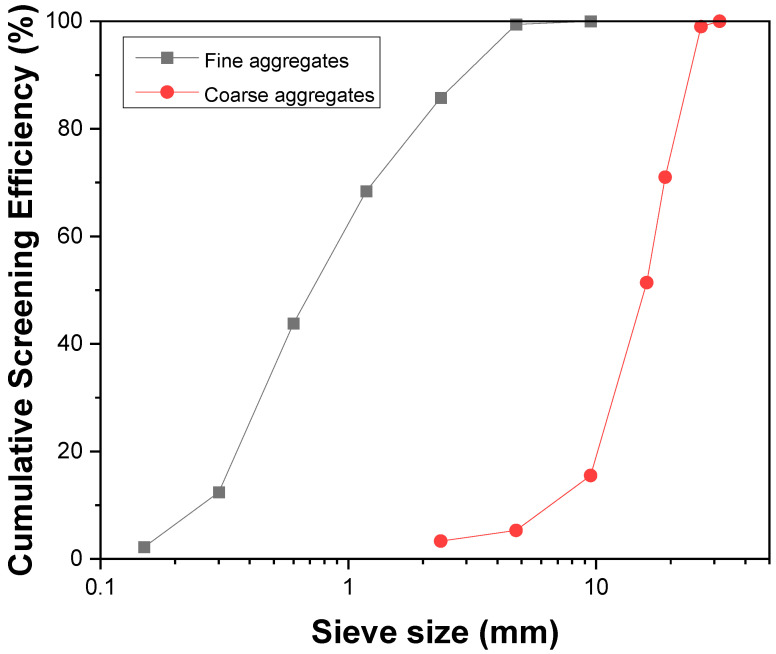
The grading curves of fine aggregates and coarse aggregates.

**Figure 2 materials-16-02905-f002:**
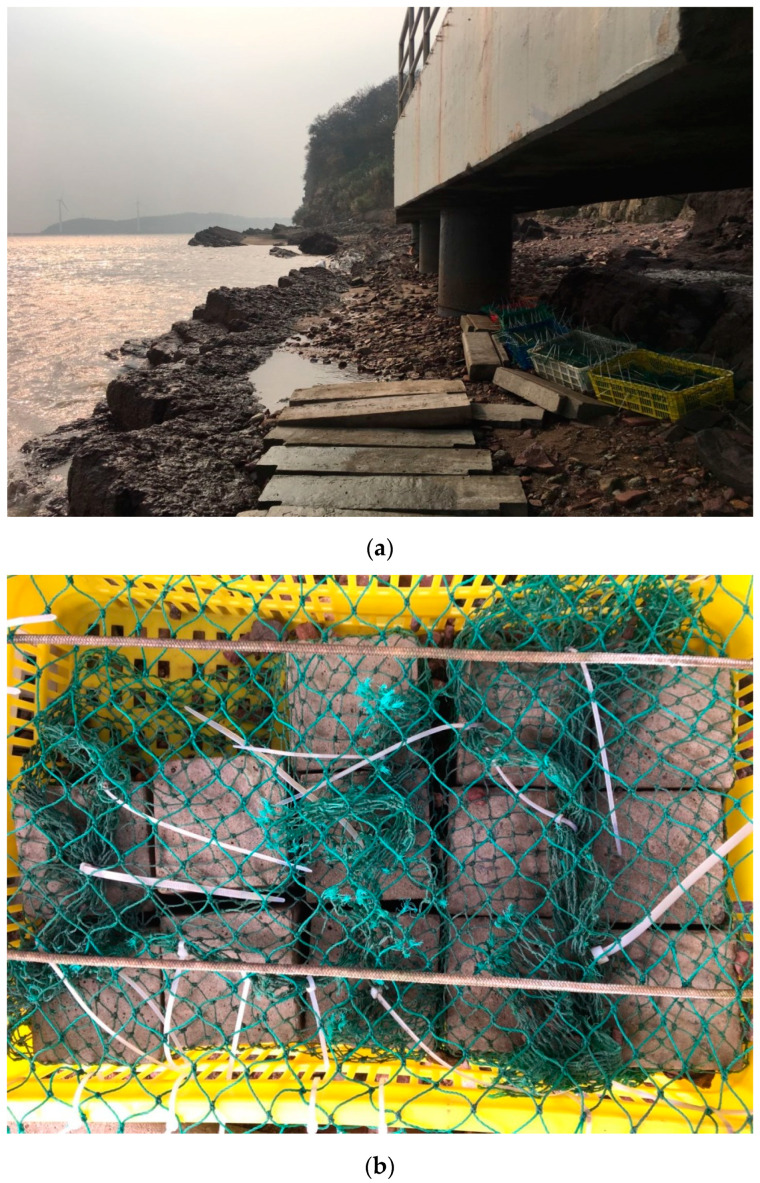
(**a**) Marine tidal zone in Zhairuoshan island. (**b**) Detail of the placed concrete samples.

**Figure 3 materials-16-02905-f003:**
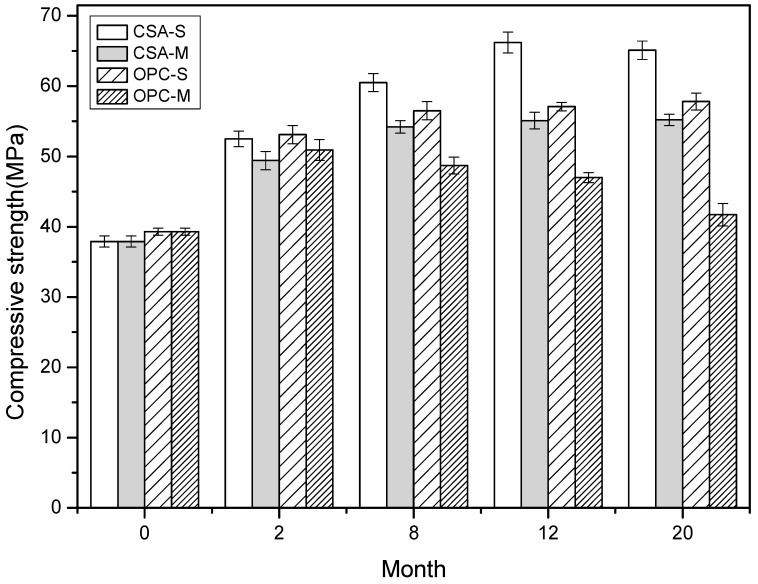
Effects of marine tidal environment on the compressive strength of CSA and OPC cement (“S” and “M” represent standard curing and marine tidal environments).

**Figure 4 materials-16-02905-f004:**
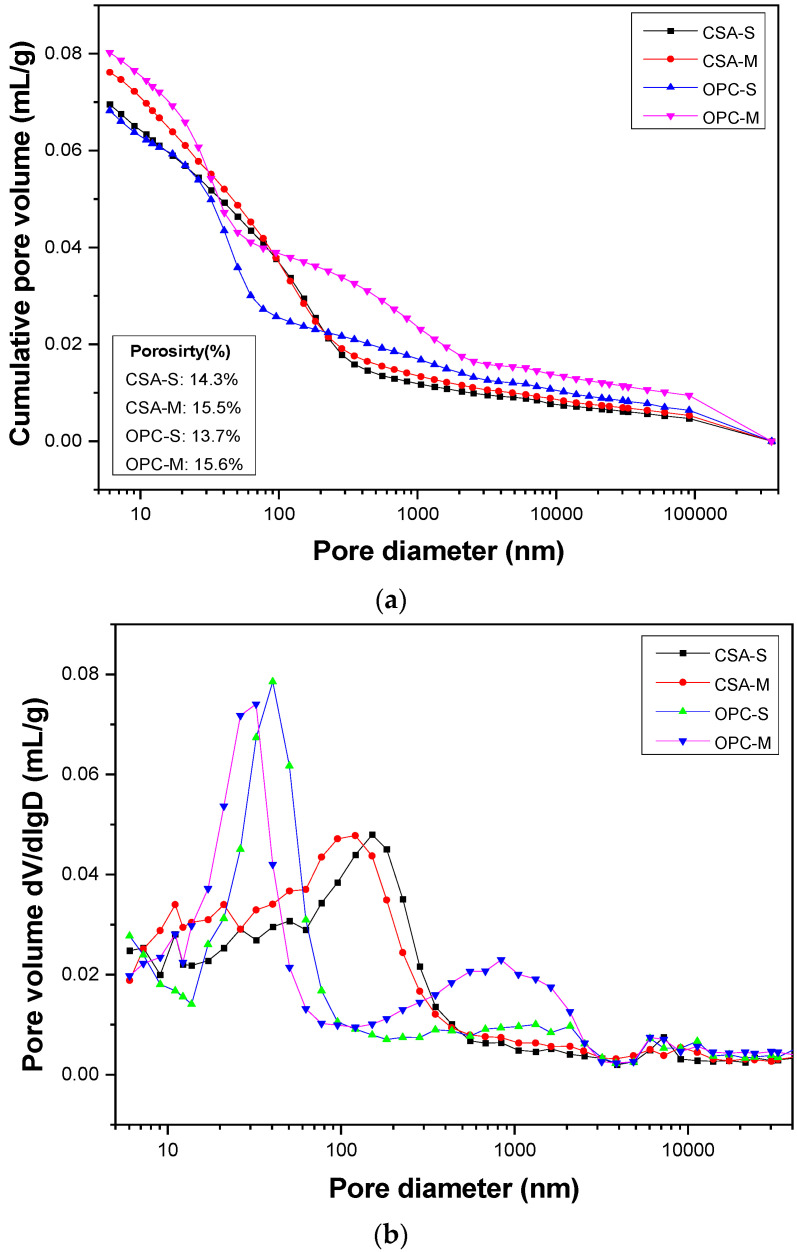
(**a**) Cumulative pore volume and (**b**) Pore size distribution (PSD) of selected samples after 2 months’ marine tidal exposure.

**Figure 5 materials-16-02905-f005:**
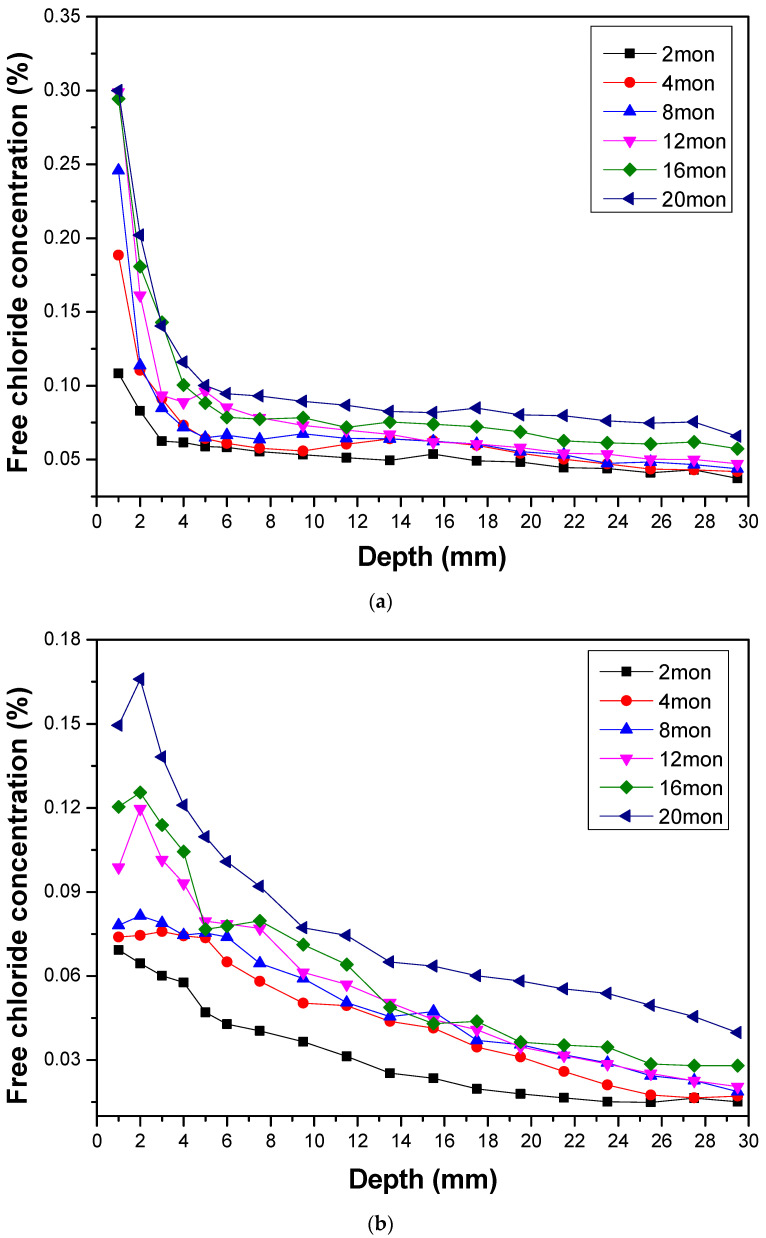
(**a**) Chloride profiles of CSA concrete samples subjected to the marine tidal environment. (**b**) Chloride profiles of OPC concrete samples subjected to the marine tidal environment.

**Figure 6 materials-16-02905-f006:**
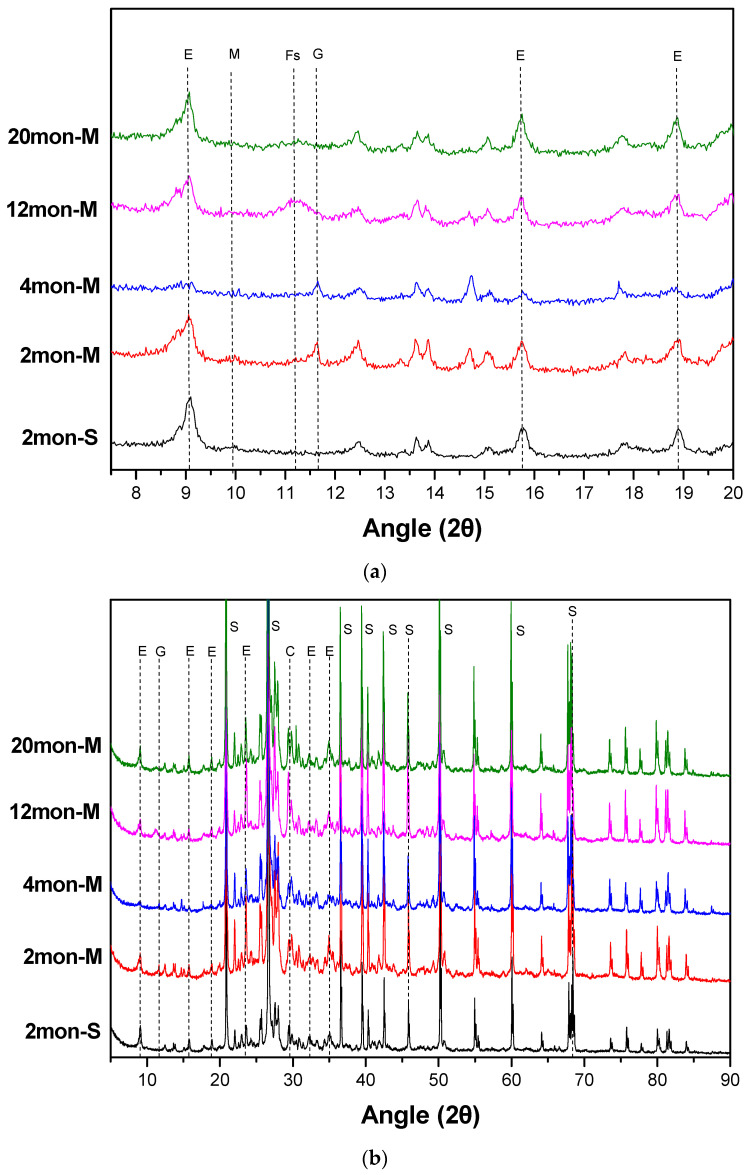
XRD patterns of CSA concrete samples after 2, 4, 12, and 20 months of marine tidal exposure. (**a**) 2θ = (5–20°), where (E)—Ettringite; (M)—AFm; (Fs)—Fridel’s salt; (G)—Gypsum. (**b**) 2θ = (5–89°), where (E)—Ettringite; (G)—Gypsum; (C)—Calcite; (S)—SiO_2_).

**Figure 7 materials-16-02905-f007:**
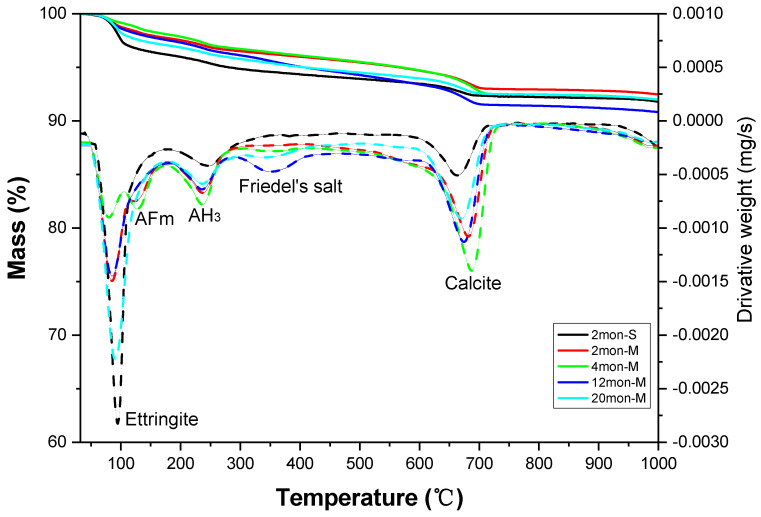
TGA patterns of CSA concrete samples after 2, 4, 12, and 20 months of marine tidal exposure.

**Table 1 materials-16-02905-t001:** Chemical compositions of CSA and OPC cement (wt.%).

	CaO	Al_2_O_3_	Fe_2_O_3_	SiO_2_	MgO	SO_3_	Na_2_O	K_2_O	Loss
CSA	44.87	25.02	3.09	9.44	3.03	12.07	0.10	0.28	1.46
OPC	58.51	5.38	4.17	22.39	4.02	1.53	0.26	0.39	1.37

**Table 2 materials-16-02905-t002:** Mix formulation to prepare CSA and OPC concretes (Unit: kg/m^3^).

Cements	w/c	Water	Cement	Sand	Stone	Superplaticizer	Retarder
CSA	0.55	169.5	308.0	807.5	1115.0	0.616	0.616
OPC	0.55	169.5	308.0	807.5	1115.0	0.616	-

**Table 3 materials-16-02905-t003:** Atmospheric and water environment of Zhairuoshan Island.

Climate	Subtropical Monsoon Climate
The highest altitude	215 m
Annual mean temperature	16 °C
Annual mean humidity	80.3%
Mean temperature of water	17.4 °C
Dissolved oxygen concentrations	7.7 mg/L
Salinity	2.5%
pH	8.1
Tide type	Irregular semi-diurnal tide

**Table 4 materials-16-02905-t004:** Ion content of seawater in Zhairuoshan Island.

Ion Type	K^+^	Ca^2+^	Na^+^	Mg^2+^	Cl^−^	SO^4−^	Br^−^	F^−^
Content (g/L)	0.519	0.073	7.957	0.856	13.600	1.980	0.011	0.007

**Table 5 materials-16-02905-t005:** Peak intensity and Full width at half maximum (FwHM) of hydration products formed in samples after marine tidal exposure via the XRD patterns.

	2θ = 9.1° (Ettringite)		2θ = 11.2° (Friedel’s Salt)		2θ = 11.7° (Gypsum)	
	Peak Intensity	FwHM	Peak Intensity	FwHM	Peak Intensity	FwHM
2mon-S	480	0.253	-	-	-	-
2mon-M	415	0.354	-	-	209	0.345
4mon-M	95	0.327	-	-	151	0.172
12mon-M	341	0.304	154	0.516	-	-
20mon-M	460	0.250	57	0.326	-	-

**Table 6 materials-16-02905-t006:** Mass losses of hydration phases formed in the selected samples after marine tidal exposure.

		Mass Loss (wt.%)			
	Ettringite (80–110 °C)	AFm (130–140 °C)	AH_3_ (220–270 °C)	Friedel’s Salt (300–380 °C)	Calcite (600–700 °C)
2mon-S	2.05	0.13	0.65	0.40	1.15
2mon-M	0.87	0.16	0.63	0.41	1.60
4mon-M	0.51	0.19	0.71	0.49	1.95
12mon-M	0.96	0.14	0.69	0.87	1.85
20mon-M	1.45	0.13	0.61	0.61	1.41

## Data Availability

Data is unavailable due to privacy or ethical restrictions.

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
