# Peer review of "Mechanical Performance and Chloride Penetration of Calcium Sulfoaluminate Concrete in Marine Tidal Zone"

_materials, 2023, doi:10.3390/ma16072905_

Round 1

Reviewer 1 Report

comments attached

Reviewer 2 Report

The manuscript presents a very interesting topic that deserves further investigation. Durability is a very important topic. However, it is also very important to investigate durability in cements with a lower co2 footprint. Exposure in a real environment is viewed very positively.

The problem of protecting steel reinforcement in an alkaline environment is correctly stated in Chapter 1. Corrosion protection of steel reinforcement is partly discussed. It would be a good way to consider and monitor the change in pH values over time for the protection of the reinforcement. Corrosion protection of steel reinforcement is discussed. When using CSA, it would be appropriate to comment on the passivation of the steel reinforcement due to the alkaline environment.

In Chapter 2, I would recommend adding a maximum aggregate grain or fraction size for sand and coarse aggregate.

In Table 2, I recommend replacing the kg·m-3 unit format with a kg/m3 style format. Consolidate throughout the document into one style. Somewhere the units are with a slash somewhere with a negative index.

I have a question about chapter 2.4.3., how many samples were used to determine one porosity value? Was the sampling from one or more locations? In general, quantities are not given, except for compressive strength (3 samples). It would be good to know for the readers.

I would add the total number of drying-wetting cycles exposed in the marina for each monitored period.

It would be good to know for an idea, I think.
